# A New Auto-Regressive Multi-Variable Modified Auto-Encoder for Multivariate Time-Series Prediction: A Case Study with Application to COVID-19 Pandemics

**DOI:** 10.3390/ijerph21040497

**Published:** 2024-04-18

**Authors:** Emerson Vilar de Oliveira, Dunfrey Pires Aragão, Luiz Marcos Garcia Gonçalves

**Affiliations:** Department of Computer Engineering and Automation, Federal University of Rio Grande do Norte, Av. Salgado Filho, 3000, Campus Universitário, Lagoa Nova, Natal 59078-970, RN, Brazil; evilaroliveira@gmail.com (E.V.d.O.); dunfrey.aragao.073@ufrn.edu.br (D.P.A.)

**Keywords:** auto-encoder, SARS-CoV-2, forecast, time-series

## Abstract

The SARS-CoV-2 global pandemic prompted governments, institutions, and researchers to investigate its impact, developing strategies based on general indicators to make the most precise predictions possible. Approaches based on epidemiological models were used but the outcomes demonstrated forecasting with uncertainty due to insufficient or missing data. Besides the lack of data, machine-learning models including random forest, support vector regression, LSTM, Auto-encoders, and traditional time-series models such as Prophet and ARIMA were employed in the task, achieving remarkable results with limited effectiveness. Some of these methodologies have precision constraints in dealing with multi-variable inputs, which are important for problems like pandemics that require short and long-term forecasting. Given the under-supply in this scenario, we propose a novel approach for time-series prediction based on stacking auto-encoder structures using three variations of the same model for the training step and weight adjustment to evaluate its forecasting performance. We conducted comparison experiments with previously published data on COVID-19 cases, deaths, temperature, humidity, and air quality index (AQI) in São Paulo City, Brazil. Additionally, we used the percentage of COVID-19 cases from the top ten affected countries worldwide until May 4th, 2020. The results show 80.7% and 10.3% decrease in RMSE to entire and test data over the distribution of 50 trial-trained models, respectively, compared to the first experiment comparison. Also, model type#3 achieved 4th better overall ranking performance, overcoming the NBEATS, Prophet, and Glounts time-series models in the second experiment comparison. This model shows promising forecast capacity and versatility across different input dataset lengths, making it a prominent forecasting model for time-series tasks.

## 1. Introduction

COVID-19 outbreak was first reported in December 2019 and resulted in significant losses requiring several non-pharmaceutical interventions [1] to its initial control [2,3]. On 30th January 2020, the World Health Organization (WHO) declared COVID-19 as an international public health emergency. Following that, some places imposed lockdown and suspension of all public transportation, flights, and trains. Despite this, the number of confirmed cases increased rapidly, and a large number of deaths were reported [2,4]. In order to help the disease control, many researchers around the world have proposed a variety of estimation techniques to predict such numbers or even a modeling on how the viruses spread out [5,6,7]. However, a global pandemic problem may be exacerbated by several factors that are not immediately apparent, such as climatic conditions and social and also political factors [8]. As a real-world problem, several factors can directly or indirectly affect the behaviors of the pandemic. Actually, studies are still trying to find the factors that really influence the spread of the virus SARS-Cov-2. Air quality, social distancing, and the use of specific equipment, among others, are known to be some of the factors that can have an impact. Researchers are not yet confident which of these factors influence to a greater degree, but there is already some good guidance as to which ones can stimulate the spread of the virus [9,10,11,12,13].

At the early stages of the pandemic outbreak, models such as SIR [5,14,15], SEIR [16,17], SEIRD [18] and ARIMA [19] were firstly used aiming at overall understanding its behavior. These methods are based on statistical and or mathematical models and they use epidemiological parameters, which are very helpful in understanding the pandemic dynamics in the short term. However, determining the long-term dynamics is a more challenging task that these models type sometimes falls short of. In this context of developing methodologies that aim to understand the consequences, behavior, and more accurate performance on long-term (or short-term too) prediction data of the COVID-19 pandemic, other studies started developing methods based on Artificial Intelligence for the prediction task. Machine learning techniques and data driven models [20] were mostly applied to predict the number of infected and deaths, besides the virus spread rate. Data-driven approaches can use artificial intelligence techniques to operate on data in order to better understand it, generating functional tools for a given action and making future estimates from time series data. Time series is how pandemic data global spreads are understood once there are temporal dependencies in its values [21]. Finally, these models might be able to assimilate complex information, which can be helpful for possible decision-making. Among the most common AI techniques used in data-driven models are artificial neural networks (ANN) [22,23,24,25]. These networks prove to be strong allies to data as sources of information, due to the ability of these networks to approximate complex functions, which usually model problems considered to be real-world [26].

Among the data-driven neural network-based models, [21] initially proposed a stacked auto-encoder architecture to univariate COVID-19 daily cases peak in some Brazilian states during the early pandemic outbreak. The results show a significant ability to predict the proposed data, but the study does not propose or implement any multivariable approach. Furthermore, due to the limited data available at that time, the authors do not evaluate the model when there is a significantly larger amount of reliable data. To evaluate the first proposed model [21] in a more complex and real-world scenario, as well as to fill a lecture gap of multiple objective models, we propose here an extend-based model for multi-variable COVID-19-related time-series. This proposed model relies on three modifications concerning the original and three other variants at the training step. As well, the new model demonstrates capability for time series forecasting. In practice, the approach provides an accurate and input-flexible forecasting model. This is a benefit in scenarios where government authorities can take the edge of precise predictions and take countermeasures to mitigate potential pandemic spread.

## 2. Related Works on COVID-19 Pandemic

Due to the demand for knowledge on the pandemic behaviors of the world population, data-driven models, more specifically statistical and machine learning techniques, were initially proposed and used to understand the series provided by COVID-19. In this direction, short-term forecasting has been provided, for example, using methods derived from smoothing model families [27]. In this case, non-seasonal multiplicative error and a multiplicative exponential trend are used in a work that aims to predict global confirmed cases and deaths related to the coronavirus over four months with a ten days step in an iterative process. The authors have compared six machine learning approaches, named Cubist Regression (CUBIST), Random Forest (RF), Ridge Regression (RIDGE), Support Vector Regression (SVR), Stacking-Ensemble learning (SEL), and ARIMA. Experiments forecast one, three, and six days ahead of COVID-19 cumulative confirmed cases in ten Brazilian states, showing SVR and SEL were the overall best models. The Eigen-Values Decomposition Hankel Matrix is applied to decompose the non-stationary data series, which results in subsequent, monocomponent subseries. Also, the unit-root tests are used to check the decomposition stationarity, assisting the ARIMA method in forecasting daily new cases for India, the USA, and Brazil. Another work in this direction uses the ARIMA model and the Prophet time-series forecasting to predict the COVID-19 spread for cumulative cases on the world total cases and on the ten most affected countries, showing the more effectiveness of the ARIMA model [28]. From the above works, we can notice that the ARIMA model is frequently used to forecast time series due to its simplicity and composition, which was originally created to deal with this kind of problem. In a relative direction, more recently, some methodologies propose a *forecast by analogy* based on previously observed behavior peaks and declines of pandemic cases in Italy [29]. The authors looked up two winter periods of 2022/2023 and 2023/2024 from COVID-19 infection. They extracted values of the weekly deaths and then computed the percentage of the change for each consecutive pair of values of the growth/decline rates of the series, comparing the most recent value in the series with the previous one. In conclusion, the authors claim the proposal is adequate and innovative, having a low percentage range of error.

Regarding another category based on machine learning models, we can find works that use artificial neural networks to solve the time series forecasting related to the COVID-19 pandemic. Ref. [30] proposes an LSTM model to bi-weekly predicting the number of confirmed cases in Canada and Italy until March 2020, also comparing the growth rate. [31] also uses an LSTM to predict the number of confirmed and recovered cases 30 days ahead of an Indian dataset. As well, [8] use a Stacked LSTM as the network architecture for both the multivariate and univariate model. Ref. [23] use deep LSTM, Stacked LSTM, Bi-Directional LSTM, and Convolutional (CNN) LSTM to 30 days forecasting, predicting, and comparing the confirmed and deaths cases of COVID-19 in India and the USA, and the authors suggested a better performance of the ConvLSTM. Ref. [32] propose an RNN to predict 30 and 40 days to daily Brazilian confirmed cases, suggesting preventive measures according to the predicted values. Ref. [33] use a Multi-Layer Perceptron (MLP) to predict the spread of pandemic world-wild to 30 days ahead, achieving a high $R^{2}$ score to the confirmed cases.

Regarding the direct application to COVID-19 using Auto-Encoders, we can also find some works related to solving the time series pandemic forecasting problem. Ref. [34] provide a comparative study with five different models of neural networks, including the RNN, LSTM, Bi-LSTM, GRU, and the Variational Auto-encoder. According to the authors, the last one achieved better results than the other models for predicting the number of new COVID-19 cases and of recovered cases. They propose to mix a Variational Auto-Encoder with LSTM networks, dividing the model into two branches. The first is a Self-Attention Auto-encoder LSTM, which receives data by daily and country virus propagation, government policies adopted, and urban characteristics. They also divide this first component into a self-attention LSTM Sequence Encoder and an LSTM Sequence Decoder. According to the authors, the self-attention mechanism makes the LSTM capable of understanding the representation of its inputs relating to the positioning of each sequence. The second branch of the model is a Variational Auto-Encoder (VAE) working in parallel with the self-attention LSTM mechanism. The VAE receives as data a dimensional, spatial matrix that has been repeated throughout the training duration with timestamps referring to the date and time. The outputs of the two model branches are concatenated in the feature dimension and sent to the LSTM sequence decoder, which returns the prediction values. This study attempts to predict the spread (accumulated number of cases) of COVID-19 worldwide and for each country separately. They trained two models, where the first forecasts three-time samples and the second predicts ten. The data used for training the models were collected from several world datasets and also for each country separately. Some countries tested were the USA, Italy, and Spain, among others. The RMSE and MSLE are used as evaluation metrics.

An integration between a deep convolutional network and an LSTM is proposed by [35]. This study uses images from the Lung Ultrasound Database (LUDB) as input, trying for a classification of these images to predict the risk of severe symptoms of COVID in the patient. Initially, CNN weights pass through to an Auto-Encoder for noise removal and feature extraction, another commonly used case for Auto-Encoders. A point to be mentioned is that the authors applied the feature extraction on the weights resulting from the CNN block. These values are presented to the LSTM network for the image classification into four patient condition scores. The model has been tested with data from the Italian COVID-19 LUDB and compared with another model named DenseNet-201. They extracted four classification metrics to evaluate the model: accuracy, sensitivity, specificity, and F1 score.

Finally, using images to make predictions and classifications, [36] applied Convolutional Auto-Encoders to detect anomalies in chest radiographs of healthy patients. The authors noted abnormal features on the X-ray of patients infected with the COVID-19 virus. Thus, they formulated a problem with only one classification class, characterizing an anomaly detection situation. This Convolutional model relies on two types of images as inputs. The inputs are only images of healthy adults and healthy adults with pneumonia other than COVID-19. By doing so, a case of detection of COVID-19 is an anomaly. The authors used other machine learning techniques joint with Convolutional Auto-Encoders, such as batch-normalization and upsampling layers. The dataset used for the experiments is the COVIDx, composed of 8851 healthy X-ray images, 6052 other pneumonia types than COVID-19, and 498 samples of COVID-19 infected people. Finally, the area under the curve calculation (AUC) ROC is used to evaluate the model and the proposed methodology.

As seen in these examples, there are several works in the literature focused on the problem of the COVID-19 pandemic. These works vary from applications that use methods dedicated to dealing with time series, and more general as artificial neural networks. Furthermore, the auto-encoder networks are not only used for time series prediction but also for detecting possible cases.

## 3. Auto-Regressive Multi-Variable Modified Auto-Encoder

We propose a multivariate input data-driven model to predict *h* samples applied for time-series forecasting tasks such as COVID-19 deaths. The proposed model in this work is a stacked auto-encoder that enhances the baseline model initially presented by [21]. Thus, to a better understanding of our model, some background theory in this topic will be presented in the next subsections.

### 3.1. Auto-Encoders

The concept of auto-encoder was first proposed by Yann LeCun in his doctoral thesis [37]. An auto-encoder is a specific artificial neural network trained to be able to copy its input into its output [38]. This structure is basically composed of two parts: the encoder and the decoder. These, in turn, can be seen as two functions Z=h(X) and $\hat{X}=g\left(Z\right)$. The first one is responsible for mapping the input data *x* to the latent space (or feature space). While the second produces the data reconstruction, mapping *Z* from the latent space back to the input data space *x* [38,39]. Figure 1 shows a high-level representation of an Auto-Encoder.

Nowadays, auto-encoders have generalized the idea of encoder and decoder in addition to the deterministic functions shown above, for mapping into stochastic functions $p_{encoder}\left(Z\;|\;X\right)$ and $p_{decoder}\left(\hat{X}\;|\;Z\right)$, where $\hat{X}$ is the reconstruction of the input signal *X*. Given real applications, it is not entirely an interest that the auto-encoder learns just how to copy the input *X*. Then, restrictions are made so that auto-encoders learn to copy the inputs roughly [39]. Furthermore, these structures are widely used for input dimensionality reduction and features extraction [38].

In the application of an auto-encoder on a given training set $T_{s}=\left\{X_{i}|X_{i}\in R^{d}\right\}$, where 1<i<n and $X_{i}$ is the *i*-th feature, this can be modeled as:(1)$$AE=\left\{\begin{array}{c} Z=h(w_{e},b_{e};X)\\ \hat{X}=g\left(w_{d},b_{d};Z\right) \end{array}\right.$$

Having the Equation (Equation 1), h(·) is the encoder and g(·) is the decoder, which are usually artificial neural networks. Terms $w_{e}$ and $b_{e}$ are the encoder parameters and $w_{d}$ and $b_{d}$ are the decoder parameters. In the case of neural networks, these parameters are the weights and biases of the network’s encoder and decoder, respectively. Training an auto-encoder is done by solving an optimization problem, in this case, to minimize the loss function given by Equation (Equation 2):(2)$$J\left(\theta \right)=\frac{1}{N}\sum_{i=1}^{n}{∥X_{i}-{\hat{X}}_{i}∥}_{2}^{2}$$
where $\theta =(w_{e},b_{e};w_{d},b_{d})$. In other words, the loss function *J* returns an error between the input feature $X_{i}$ and the output of the network ${\hat{X}}_{i}$, and another algorithm uses this error to adjust the weights and bias values. For the optimization solution present in Equation (Equation 2), the Gradient Descending or Stochastic Descending algorithm is normally used.

Throughout time, several types of auto-encoders have been proposed by different researchers. The Sparse Auto-encoder (SAE) is derived from the original auto-encoder, applying sparse regularization in the latent space. The Denoising Auto-encoder (DAE) is an auto-encoder that removes the noise from the input data. The Convolutional Auto-Encoder (CAE) extends from the conventional auto-encoder by instantiating the encoder and decoder function with convolutional neural networks [38]. The Contractive Auto-encoder, Variational Auto-Encoder, and Modified Auto-Encoder [21] are other types of auto-encoders that are present in the literature.

### 3.2. Modified Auto-Encoder

Due to the short data size during the early COVID-19 outbreak, the authors proposed an Auto-encoder based model for uni-variable forecasting daily cases, in their baseline approach [21]. The model has one single input layer followed by three encoder layers that receive the same input data, where each encoder generates its own latent space. The three encoder latent spaces are concatenated before they are shown to the prediction layer. At the same time, each decoder generates its decoder output. The three decoded outputs have their average calculated to dimension match the input to error calculation. Finally, the prediction layer output has its values approximated to the desired value. Figure 2 shows a layer schematic representation of the originally proposed modified auto-encoder.

In this work, we propose three modifications and extended variations for the above Modified Auto-Encoder model, which was originally presented by [21]. We base these three model types on three structural changes through which the model perceives the inputs, on the manner it manages the latent space, and on the approach to adjusting the model weights.

### 3.3. Baseline Structural Changes

The changes in the adopted model of the Modified Auto-Encoder and the results of this work are based on three basic conditions. The first in how the model observes the inputs. The second is how the model deals with the latent space. And the third is how to fit the model in the training steps. The first proposed change is to add a new Auto-Encoder for each input feature. Figure 3 shows the model flow overview and how our proposed base model deals with *N* time-series features as input.

A new auto-encoder network is created for each series (feature), thus the whole structure has the same number of encoded series. Following the original modified auto-encoder architecture, increasing the input features of the model would require at least three auto-encoders for each input series, which can lead to a large model. To mitigate this potential problem and keep the network simple. We use only one auto-encoder per input feature and more neurons at the hidden layer, thus, denominating it as *growth layer*. Like the three auto-encoder latent spaces from the first proposed network, this aims to amplify the network information extraction. Figure 4 shows a representation of this new proposed auto-encoder architecture.

From Figure 4 we can see that the architecture of each auto-encoder can vary. But, there must always be a layer of greater or equal size between the input and the last encoder layer, equals to the decoder architecture. A way to represent the architecture of this auto-encoder is [IN,G,LS] to the encoder and [LS,G,OT] to the decoder. IN is the input size, *G* is the growth layer size, and LS is the latent space size. The OT is the output layer size, which always has the same size as IN. We also change from the LSTM network to MLP fully connected layers. From this point, it is possible to understand the second proposed modification. As in the original proposal, all of those encoded values are unified (concatenated) into a single latent-space series. The difference is that we concatenate latent spaces from different series of inputs instead of the same one.

From Figure 3, we can observe that the latent space has a reduced representation of each input time series, which are called LS1,LS2,…,LSN. This final concatenated latent space undergoes one more linear transformation (prediction layer) to return the forecasting values. The training step has a fundamental importance to understanding on how the model works. As explained in Section 3.1, the model tries to minimize the error between the predicted value and the expected one. In our model, this occurs when each auto-encoder tries to recreate the input of the series (RC1,RC2,…,RCN and for the prediction error. We summed the auto-encoder’s recreation errors into a single error named $ae_{error}$. The first equality in Equations (Equation 3) shows how $ae_{error}$ is calculated with terms $e_{1},e_{2},\cdots ,e_{n}$ being the reconstruction errors of each auto-encoder in the model. We named the loss function Loss, so the prediction error $pd_{error}$ is the result of this function from the forecasted value and the expected value. The set of Equation (Equation 3) show how these errors and the model final $t_{error}$ are acquired.
(3)$$\begin{array}{cc} ae_{error} & =\sum_{i=1}^{n}e_{i}\\ pd_{error} & =L(forecasted\;|\;expected)\\ f_{error} & =ae_{error}+pd_{error} \end{array}$$

The set of Equation (Equation 3) implements our third change. These are the proposed fundamental changes compared to the original architecture of the model on the COVID-19-related time series prediction problem. Having the above major changes been implemented, we now introduce other three minor modifications, which are done at the training steps and weights adjustment. This lead us to a three proposed models variations, that aim to evaluate the whole model’s flexibility and, as expected, to improve prediction accuracy.

### 3.4. Model Type#1

The model titled as Type#1 is the one explained in Section 3.3 without any modification. This model type calculates all errors and adjusts the weights to reduce the accumulated auto-encoder and predictor errors at a glance. For each step in training, the auto-encoders have errors in recreating their feature input. Also, the predictor layer has an error concerning the target prediction values. The sum of the recreation auto-encoder and the predictor errors generates the total model error. The backpropagation algorithm receives these errors and calculates the fitted weights. These steps are described in Algorithm 1.    
**Algorithm 1:** Train loop for model Type#1
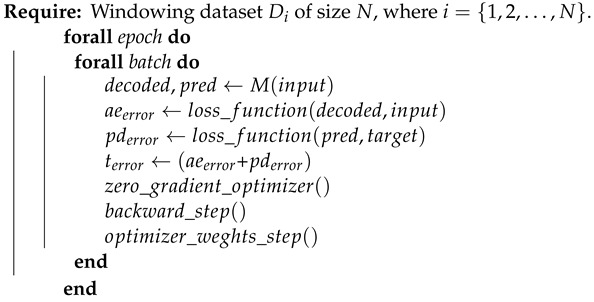


### 3.5. Model Type#2

The model Type#2 is the one explained in Section 3.3, but we previously fit all the auto-encoders, and then the predictor is trained after this. For each input time-series feature, we have the auto-encoder reconstruction error singly. We freeze the decoder component weights and fine-tune the encoder component weights and the predictor error at a glance. Algorithm 2 describes the training step for this auto-encoder type.    
**Algorithm 2:** Train loop for model Type#2
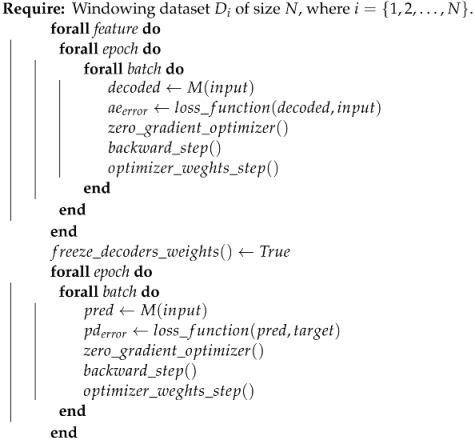


    From Algorithm 2, we can notice that the encoder component is not based in the loss error on the predictor train, but with their weights not freezing, the backward gradient function also calculates it and fine-adjusts it.

### 3.6. Model Type#3

The model named as Type#3 is the one explained in Section 3.3. However, in the same way as it occurs in the Type#2, we fit all auto-encoders previously and then the predictor is trained after this. For each input time-series feature, we have the auto-encoder reconstruction error singly. We freeze the decoder component weights, and for this type, we also freeze the encoder component weights. Algorithm 3 shows the training steps for this type.

From Algorithm 3, we can notice that the auto-encoders components have their weights freezing, so the backward gradient function calculates only the predictor’s new weights and adjusts them.

These three variations intend to discover whether training and weight adjustments of the predictor network separately from the auto-encoders can improve or worsen the network’s performance, as it will be shown in the experiments, next. By evaluating this point and having this information, it is possible to make the auto-encoders independent, training them in isolation to serve only as “boxes” that, when receiving input from a series in which they have already perceived their nuances and variations (in this case has been trained), they are able to generate a latent space that has a good representation of the input. At this point, the adjustment step split is relevant to understanding how much the predictor’s error can influence the weight adjustments and consequently influences the model when coding the inputs.    
**Algorithm 3:** Train loop for model Type#3
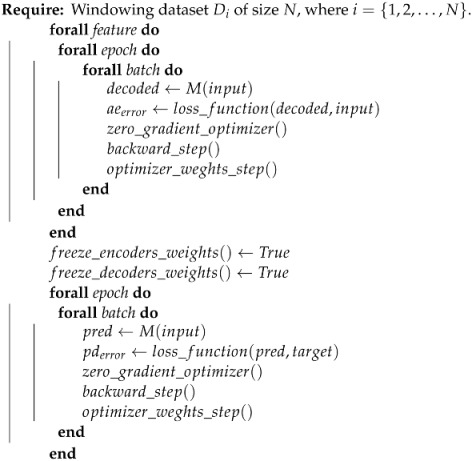


## 4. Experiments

To evaluate the proposed model in the time series forecasting task for COVID-19-related data, we promote a comparison of the three proposed model types with the approaches proposed in two previous works. First, [8] has proposed a Long Short Term Memory (LSTM) model to forecast the number of COVID-19 deaths for the São Paulo state, in Brazil. To evaluate the model, the authors used a multi-variable dataset from 27 March 2020, to 3 June, 2021. The dataset was composed of two directly related time series of the pandemic and three indirect (or possible) related time series. The two directly related time series were the number of COVID-19 cases and deaths, and the indirect series was the temperature, humidity, and the air quality index (AQI). The AQI is the max value of some measured pollutants in the air. The authors trained 50 trails from their model architecture with four different configurations of input series. From those trials, they extracted a statistical distribution concerning the model Root Mean Square Error (RMSE) from the entire and test forecasting datasets. To read more about this study, please refer to the original author’s article [8], including the implementation and dataset, which are accessed on github (https://github.com/Natalnet/ncovid-air-paper (accessed on 21 November 2023)). The comparison with this work aims to position our work in the literature concerning models with a statistical training evaluation through multi-variable input features, directly and indirectly, related to the COVID-19 pandemic. Due to a fair comparison, we used the same dataset and values for window moving average, window input size, and other parameters described in Table 1.

The moving average used to enter a temporal correlation and decrease noise on the inputs was 14 samples (biweekly). We also used a window of seven series samples for each input with an overlap of six (6) series samples from each other. Thus, the difference between each input was only the current time sample. We use the same auto-encoder architecture {7,28,7}, as the explanation in Section 3.3 (mirror to the decoder), for all three proposed model types. We use a learning rate of 0.001 for all optimizers in all trails, the RMSE as the loss function, the Rectifier Unit (ReLU) as the activation function between the neurons layers, and a batch size of 16 inputs before weights adjustment. We do not use any dropout or data normalization layer on our model trails.

As a second work to compare, [40] proposes to evaluate six time-series models for the COVID-19 forecast task. The authors compared the TBAT, ARIMA, HWAAS, NBEATS, Prophet, and Gluonts models. Also, they applied the Friedman statistical test to rank the models. This work performed the models over the ten more affected countries worldwide at the study point, the United States, Spain, Italy, the United Kingdom, France, Germany, Russia, Turkey, Brazil, and Iran. All datasets to train and forecast were composed of the percentage of active COVID-19 cases concerning the total population from the pandemic start to 4 May 2020. To see more details about these models implementation( https://github.com/ML-Upatras/COVID-19-A-comparison-of-time-series-methods-foractive-cases-forecasting (accessed on 26 November 2023)) and dataset formulation, refer to the original work [40]. We could not find any specific explanation from the authors about the architecture or hyperparameter variance of the models to attain the results performance. So, we have done a grid search in our models, changing only the seed generator to the layer’s weights values, from 1 to 100, with a step of 2 values (1,3,5,…99) over 4000 epochs each. We use the same auto-encoder architecture {7,28,7} to three proposed model types as in the past comparison experiment. The learning rate was also 0.001 for all optimizers, the RMSE as the loss function, the Rectifier Unit (ReLU) as the activation function between the neurons layers, and one-third of the training dataset as batch size. Due to the low absolute values of COVID-19 indicators percentage, we multiplied the dataset by 100 before the training. This maneuver facilitated the proposed model to recognize the variance of the values. After, we divided it by 100 to conserve the scale and extract the metric. We also split the dataset into 75% to train and 25% to test to have the same length as the comparison experiment, using a window of seven series samples for each input with an overlap of six (6) series samples from each other. Comparing the proposed model to this work aims to position our work in the literature concerning traditional time series models. Also, evaluated our model for another pandemic indicator prediction that is not directly a collected data time-series, such as deaths and cases. We selected the model with the seed generator resulted in the lowest final prediction RMSE. Table 2 shows the seed with the lowest prediction RMSE for each country.

Thus, the overall main goal of the comparing experiments is to evaluate our proposed model among different use cases, quantify its flexibility, range of applications, and if it can reach state-of-the-art COVID-19-related time-series forecasting.

## 5. Results

All implementations of the experiments presented above were executed in Python, using the machine learning framework Pytorch [41]. We also used other frameworks and libraries such as Numpy [42], Pandas [43], and Scikit-Learn [44]. In this Section, we present the results of these experiments that we performed for comparing our implementation to the other works.

### 5.1. Comparison with Stacked LSTM Uni and Multivariate

In the first work for comparison [8], the value showed and measured in the experiments to evaluate the model was the median of the RMSE metric over the distribution. The authors extracted the forecast RMSE from the entire model and the test datasets. Table 3 and Table 4 show the comparison of the three proposed model types to the referenced work, for entire and test datasets respectively.

Table 3 shows the results given by the three model types for the entire dataset. From these values, it can be noticed that our three proposed models attain better performance than the comparison model. In the first configuration, our model was more accurate at 79% for Type#1 and Type#2 and almost 91% for Type#3. These results were the most improved for median distribution values in the entire dataset. Concerning the lowest value obtained by the comparison model, to Configuration 3, our model Type#3 achieved a 75.4% improvement rate. Table 4 shows the results acquired by the three model types for the test dataset. The models of Type#1 and Type#2 did not achieve better results in any distribution trial. Although, the model Type#3 attained a 42.4% improvement to the best value of the comparison model for Configuration 1. These results lead to a reasonable accuracy for forecasting data from a set of information (Table 3). Furthermore, the proposed model Type#1 and Type#2 could not improve the performance in the test dataset, where only Type#3 achieved the lowest metric values concerning the comparison with LSTM (previous work). From these values, we notice that the proposed model Type#3 can better generalize than the other model types. Figure 5 shows the visual forecasting for the best-fitted model for each configuration.

From Figure 5, it is possible to see the daily forecast for the whole dataset. Remarks in the colored area are the samples that compose the test dataset. The first configuration presents the prediction more visually closer than the others. The Configurations 2 and 3 have a noise increase, but still with a reliable visual approximation. The last configuration shows the worst visual approximation, mainly at the curve peak and test part, where the predictions were less than the desired values.

In this experiment, the major difficulty was the time taken to train all 50 model trails of each type, for all configurations. Despite that, the reliability and dataset length contribute to the required few epochs to fit the models, contributing to not increasing, even more, the time taken.

### 5.2. Comparison with TBAT, ARIMA, HWAAS, NBEATS, Prophet, and Gluonts Models

In the second work for comparison [40], the values chosen to evaluate the models were the RMSE in a seven days window, from 28 April to 4 May. Table 5 shows the seven days test forecast of the three proposed model types and the referenced work comparison (TBAT, ARIMA, HWAAS, NBEATS, Prophet, and Gluonts models).

Table 5 shows the results acquired by the three proposed model types joined with the six referenced models. In italics, we have the lowest values for the RMSE metric concerning the proposed model types. In bold format, we show the overall lowest RMSE value by country. In Spain, Germany, and Brazil, the proposed model reached the lowest overall metric value concerning the comparison models. Type#2, Type#1, and Type#3 obtained those results for Spain, Germany, and Brazil, respectively. From these values, it is possible to notice that our three proposed models attain better performance than some models and are worse than others. We conjecture that this may be due to the smaller training data size that can affect the proposed model’s performance. Table 6 shows the Friedman statistical ranking recalculation for the comparison models and the proposed.

Table 6 shows the recalculated rank for comparison and proposed models. It is possible to see the model Type#3 and Type#1 have better overall Friedman statistical test rank than the NBEATS, Prophet, and Gluonts models. Model Type#2 was the worst performed among the proposed models for this experiment, staying better only than the Prophet and Gluonts models. This rank shows us an overall model classification related to all acquired metrics in all countries. Those results can lead us to think that our proposed model has affordable results compared to the well know time series-focused methods.

In this experiment, the difficulty regards the short dataset length, requiring several epochs to fit the models. Furthermore, as we did not find any information about the comparison model hyperparameters, we could not have a north to guide our grid search. That can cover some aspects which turn the comparison not equitably as we desired. As a good point, the Friedman statistical test ranking was a functional method to know the overall position of each model.

## 6. Discussion

The experiments and results with the proposed model have proven its importance in researching time series particularly those scenarios requiring feature scalability. However, because these are mathematical models, it is important to notice that we are discussing here an approximation, not providing, thus, an exactly solution to such kind of complex problems, as it is the case with COVID-19 pandemic. There is a multifaceted issue that affect a variety of systems such as health, politics, and economics over time. Therefore, extensive work is required to collect considerable features that can be used as input data for the proposed model. To accomplish this properly, one must thoroughly interact with the environment in which the model is to be used, understanding how the training data were gathered, and, in many circumstances, adding domain expertise when available data is insufficient.

Beyond typical predictive performance, a model’s criteria must be thoroughly stated and evaluated with more complex scenarios and may outperform other models. For example, in the first comparison, we can see that the proposed model achieved reasonable results in predicting COVID-19 deaths when given a multi-variable dataset as input. Furthermore, the distribution metric values obtained show that our model is comparable or even better than the LSTM model [8]. Our proposed model absorbed temporal information from data without modifying any specific neurons. Nevertheless, two of the three proposed model types (Type#1 and Type#2) demonstrated difficulty in generalizing, acquiring worst performance than the LSTM on the test dataset.

The model’s performance is also threatened by the dimensionality of the latent spaces, which could be too low. However, the results demonstrated that this does not lead to information loss, whereas high dimensionality can result in redundancy in loss of generalization and imprecise representations with low-quality data reconstruction. The second work comparison demonstrates the proposed model also outperforms some of the specific time-series methods developed to handle prediction tasks. In some cases, the proposed model had an overall better performance, ranking fourth to Type#3, fifth to Type#1, and seventh to Type#2 among nine models. Nonetheless, the results show that our model does not perform well when dealing with short forecast data and low (percentage) input value samples. This leads us to consider and develop model improvements, such as other model architectures and grid search options, intending to reduce prediction error. On the other hand, increasing the number of parameters increases the risk of over-fitting, especially if the training data is limited. Regularization strategies, such as dropouts or L1/L2, may be necessary to mitigate this kind of issue.

In addition, improving the accuracy in forecasting requires that the assemblage of noise reduces the model’s ability to generalize. The model consists of two main stages, during which noises will be aggregated. The first method involves concatenating the latent spaces to predict COVID-19 deaths. The second phase consists not only of reconstructing input data but there is also of noise while gathering input feature sequences. As a result, there exists an increase of error in the second stage, both the built-in establishment of latent spaces and reconstructing input data. Furthermore, adding neurons to the growth layer, which is critical to the model, is effective up to a certain point, and here resides one of the most significant points to better analyze and put effort into. Despite this, the limited number of presented features forces the model to generalize with the problem’s underspecification.

We believe that while the model can capture linear relationships between data, it struggles to represent complex non-linear relationships, particularly for sets of data with non-linear patterns.

## 7. Conclusions

Scenarios such as a world pandemic require the scientific academy to play a critical role in assisting authorities and rulers in making decisions regarding the virus’s spread. Studies can be conducted to identify environments that are more prone to contamination, factors that contribute to an increase in deaths and cases, and prediction methods. These prediction methods function by observing the behavior of pandemic-related series and forecasting the increase or decrease of their indicators. In this direction, the current work proposes a novel multi-feature Auto-Encoder-based methodology for COVID-19 time series forecasting.

The proposed model comes with three types of variations, which include changes to the original model architecture, the training step, and the model weight adjustments performed. To address, verify, and evaluate the proposed model, we conducted comparisons with two other works, including an LSTM model targeting COVID-19 deaths based on uni-variable and multi-variable datasets. Also, we compare with other six time-series-based models in forecasting the percentage of active COVID-19 cases in countries’ populations.

The proposed model shows a substantial capacity for forecasting time-series-related tasks, positioning the model in the literature among multi-variable neural networks and time-series traditional models. However, the experiments also show some model limitations, such as observed in model variation Type#3 for short data length availability or multi-variability.

The findings motivate seeking improvements, such as composing the architecture of various neural network neuron types and increasing the number of layers in the auto-encoder. Furthermore, future research should investigate the proposed model’s limitations, complexity, and scalability. In addition, we intend to implement a model extension that will allow the model to assimilate new relevant time series without requiring full re-training. Finally, additional work will be done to promote a test to determine whether the internal growth of the number of neurons in the middle encoder (and decoder) layer has a positive or negative impact on model results.

## Figures and Tables

**Figure 1 ijerph-21-00497-f001:**
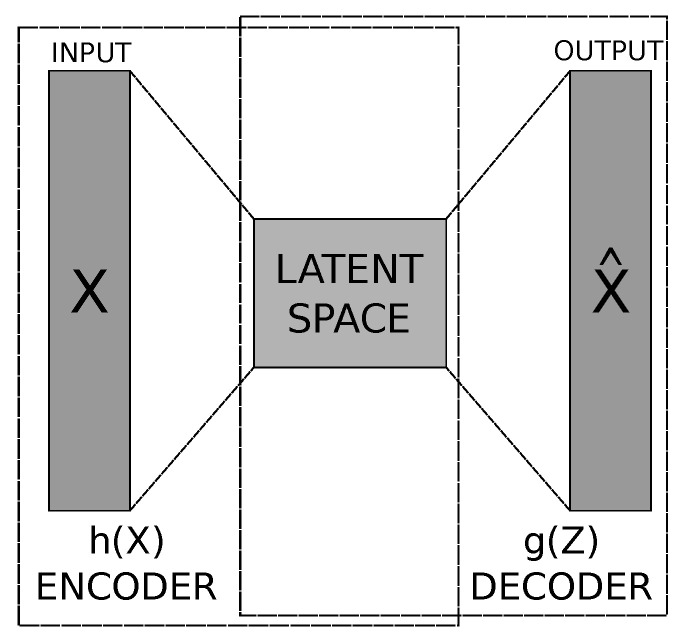
High -level representation of an auto-encoder.

**Figure 2 ijerph-21-00497-f002:**
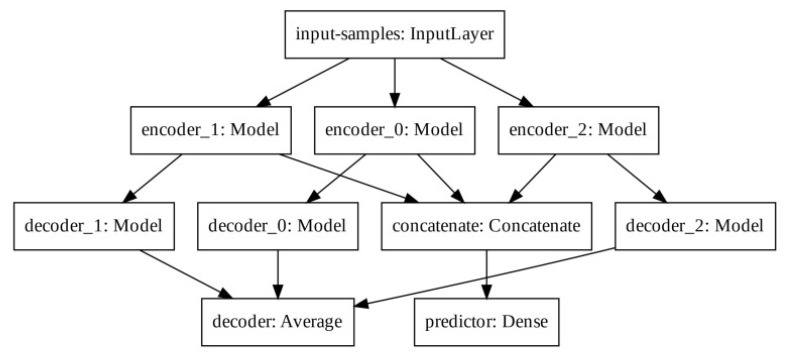
Original Modified Auto-encoder model. The image has author’s authorization to use [21].

**Figure 3 ijerph-21-00497-f003:**
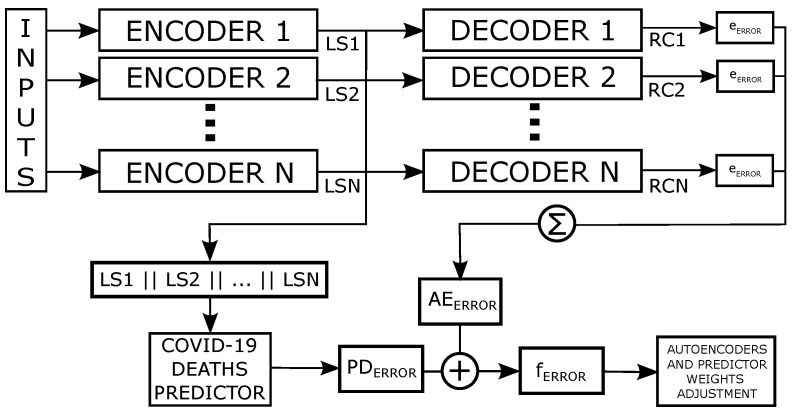
Multi-variable Modified Auto-encoder model flow overview.

**Figure 4 ijerph-21-00497-f004:**
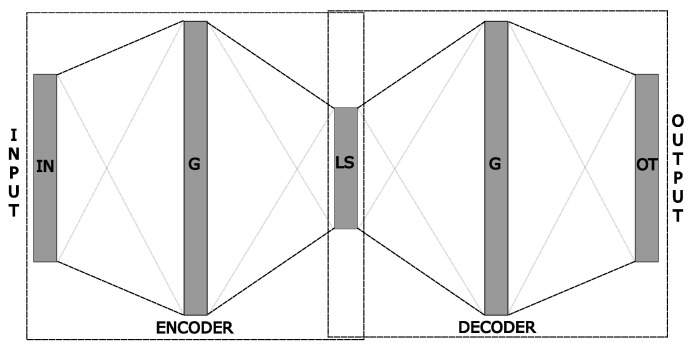
The encoded values are concatenated, forming a single latent space.

**Figure 5 ijerph-21-00497-f005:**
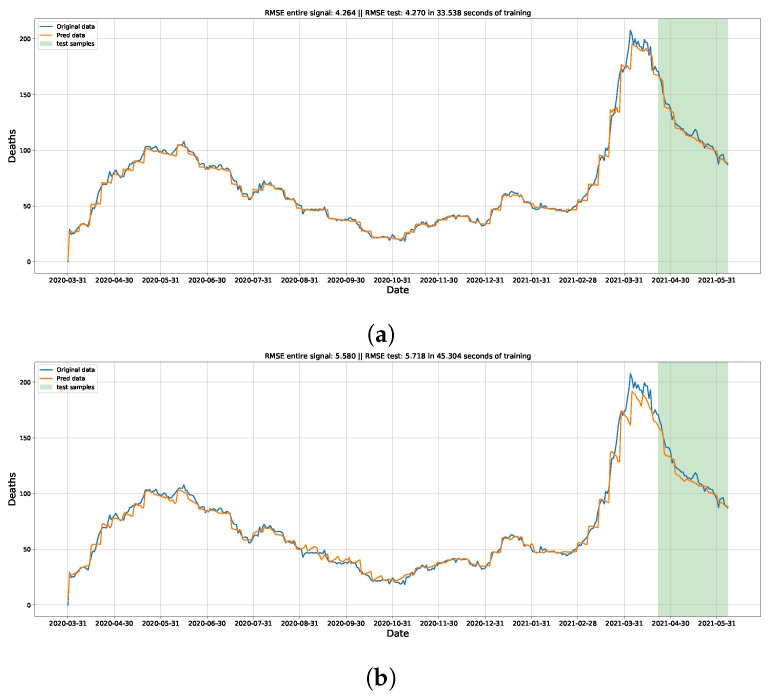
On the vertical axes of the images is the count of new daily deaths caused by COVID-19 (noncumulative). On the horizontal axes are the dates of the occurrences. The model inputs are windows of size 7 without sample overlap between them. A seven-day prediction was also made, as in the original comparison article. The blue and orange lines are the original and forecasted data, respectively. The green span area is referent to the samples in the data test set. (**a**) Visual forecasting plot from the best-fitted model to the test distribution for Configuration 1 (D). This forecasting was resulted by the model Type#3 trained with the seed value of 81. (**b**) Visual forecasting plot from the best-fitted model to the test distribution for Configuration 2 (D + A). This forecasting was resulted by the model Type#3 trained with the seed value of 89.; (**c**) Visual forecasting plot from the best-fitted model to the test distribution for Configuration 3 (D + A + T + H). This forecasting was resulted by the model Type#3 trained with the seed value of 99. (**d**) Visual forecasting plot from the best-fitted model to the test distribution for Configuration 4 (D + C + A + T + H). This forecasting was resulted by the model Type#3 trained with the seed value of 97.

**Table 1 ijerph-21-00497-t001:** Input features and some hyperparameters for Comparison 1.

Configuration	Input Features	Trails	Epochs	Train/Test
1	Deaths.	50	150	90%/10%
2	Deaths, AQI.	50	150	90%/10%
3	Deaths, AQI, Temperature, Humidity.	50	150	90%/10%
4	Deaths, Cases, AQI, Temperature, Humidity.	50	150	90%/10%

**Table 2 ijerph-21-00497-t002:** Seed generator to the layer’s weights values to each country and proposed model type.

Country	Type#1	Type#2	Type#3
US	25	25	17
Spain	57	7	29
Italy	19	29	35
UK	3	3	17
France	31	17	1
Germany	5	53	21
Russia	49	3	55
Turkey	27	25	9
Brazil	49	3	51
Iran	13	25	25

**Table 3 ijerph-21-00497-t003:** RMSE median values from the comparison and proposed models for the entire dataset.

Configuration	Type#1	Type#2	Type#3	LSTM
1	10.55 (79.2%)	10.64 (79.0%)	**4.79** (90.5%)	50.83
2	10.39 (72.9%)	10.50 (72.6%)	**6.23** (83.7%)	38.34
3	11.22 (57.0%)	11.01 (57.8%)	**6.41** (75.4%)	26.10
4	12.46 (71.9%)	11.18 (74.8%)	**8.54** (80.7%)	44.47

The lowest RMSE values acquired among the proposed model types are in bold. Between parentheses, we show the percentage improvement of the RMSE metric concerning the LSTM comparison model.

**Table 4 ijerph-21-00497-t004:** RMSE median values from the comparison and proposed models for the test dataset.

Configuration	Type#1	Type#2	Type#3	LSTM
1	13.80 (−38.4%)	14.58 (−46.2%)	**5.74** (42.4%)	9.97
2	13.41 (−10.7%)	14.05 (−16.0%)	**8.13** (32.8%)	12.11
3	15.64 (−26.1%)	15.73 (−26.8%)	**8.80** (29.0%)	12.40
4	15.59 (−5.90%)	15.39 (−4.50%)	**13.20** (10.3%)	14.72

The lowest RMSE values acquired among the proposed model types are in bold. Between parentheses, we show the percentage improvement of the RMSE metric concerning the LSTM comparison model. The negative value means no improvement, where the model obtained a worse result.

**Table 5 ijerph-21-00497-t005:** RMSE values from the comparison and proposed models for the seven-day test dataset.

Algorithm	US	Spain	Italy	UK	France
ARIMA	**0.007421**	0.080094	**0.005628**	0.005484	0.060824
Prophet	0.013877	0.065433	0.019217	0.007634	0.044482
HWAAS	0.172957	0.031497	0.006616	0.004366	0.011007
NBEATS	0.036958	0.050492	0.008645	0.037623	**0.004220**
Gluonts	0.044805	0.108842	0.043551	0.046134	0.010549
TBAT	0.009873	0.029295	0.005810	**0.004310**	0.007003
Type#1	*0.016295*	0.010258	0.008700	0.099830	0.056099
Type#2	*0.016295*	* **0.008289** *	*0.007309*	0.154313	0.075555
Type#3	0.035227	0.032317	0.015512	*0.043483*	*0.046800*
**Algorithm**	**Germany**	**Russia**	**Turkey**	**Brazil**	**Iran**
ARIMA	0.006431	**0.001536**	0.004442	0.004194	0.002628
Prophet	0.037139	0.014681	0.044595	0.009279	0.016281
HWAAS	0.004586	0.002295	**0.000887**	0.005717	0.001046
NBEATS	0.013192	0.027078	0.018265	0.010870	0.003745
Gluonts	0.057523	0.034479	0.093839	0.002836	0.002277
TBAT	0.003389	0.002193	0.001946	0.005621	**0.000425**
Type#1	* **0.002979** *	0.032710	*0.009678*	0.008168	0.008500
Type#2	0.003639	0.043008	0.010902	0.012032	0.010388
Type#3	0.008491	*0.017300*	0.010907	* **0.001963** *	*0.002387*

The lowest values for the RMSE metric concerning the proposed model types and the overall lowest RMSE value by country are showed in italics and bold, respectively.

**Table 6 ijerph-21-00497-t006:** Friedman statistical test ranking (significance level of α=0.02).

Rank	Alghrithm
2.10	TBAT
3.70	ARIMA
3.80	HWAAS
5.20	Type#3
5.25	Type#1
5.80	NBEATS
6.15	Type#2
6.30	Prophet
6.70	Gluonts

## Data Availability

https://github.com/Natalnet/ncovid-autoregressive-mae-paper.

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
