# Peer review of "A New Auto-Regressive Multi-Variable Modified Auto-Encoder for Multivariate Time-Series Prediction: A Case Study with Application to COVID-19 Pandemics"

_ijerph, 2024, doi:10.3390/ijerph21040497_

Round 1

Reviewer 1 Report

Comments and Suggestions for Authors

This paper aims to present a new auto-regressive multi-variable auto-encoder for time series prediction. 

Unfortunately, it is very difficult to provide an evaluation of the proposed technique because the topics the paper tries to address are too broad.

There are several problems:

- Section 2 includes a lot of material that is not used in the rest of the paper, for example the mathematical details of the SIR, SEIR and ARIMA models,

- Relevant bibliography is missing concerning other approaches.

-The presentation of the proposed methods is not clear enough it starts claiming that it is an extension of the work proposed by Pereira which apparently is not described.

- Which is the main goal of the experiments comparing the three models or comparing the proposed technique with other approaches?

- A good discussion section is missing.

- The impact on possible use cases is not clear.

The paper could be improved as follows:

1) Select a (restricted) goal, for example predicting deaths or new cases.

2) Provide just one background section more focused on the main goal selected.

3) Introduce the proposed method providing all the necessary details.

4) Present an experiment that focuses on the demonstrating the selected goal.

5) More specific references should be added, for example if the authors will select as a main goal predicting deaths, they will find updated references in this paper:Marco Roccetti. Drawing a parallel between the trend of confirmed COVID-19 deaths in the winters of 2022/2023 and 2023/2024 in Italy, with a prediction[J]. Mathematical Biosciences and Engineering, 2024, 21(3): 3742-3754. 

6) provide a use case which illustrates how the proposed technique ca be used in practice.

Minor points

Pag 1: line 34: mortality --> case fatality

Comments on the Quality of English Language

The use of English must be improved, proof read done by a native English speaker would be recommended.

Author Response

Please, see attached file.

Reviewer 2 Report

Comments and Suggestions for Authors

The SIR model is too old, why used?

Equation 2 is not correct; should be revised

A comparison should be made  between the SIR model and SEIR Model

Figure 8 needs some elaborations, it is not clear for readers 

Table 5. RMSE values from the comparison and proposed models for the seven-day test dataset.

it should be mentioned which model is the best

The paper is very good and deserves publication  

Comments on the Quality of English Language

English is fine

Author Response

Please, see attached file

Reviewer 3 Report

Comments and Suggestions for Authors

The authors have proposed 3 variations of a model to forecast the COVID-19
related time series with the use of the Auto-Encoder Recurrent Neural Network
Architecture. They have compared the results of 7 days forecasting with other models
available in the literature. The large differences in RMSE values obtained for
different countries with the use of different models (see Table 5)
raise questions
about the predicative capabilities of all the models, including those proposed by
the authors. In addition, in some cases ARIMA and TBAT yield better predictions
(see Table 5).
  It is impossible to analyze the results presented in Figure 8. What is “Deaths”
on the vertical axis? Do you mean the daily numbers of new deaths in Sao Paulo?
You need to specify this in the Figure or in the caption. Is it only one day forecast?
Hove many days were used for input? I think that one-day forecast has very limited
area of applications. How looks the results for 7-days forecast?
  The authors have not demonstrated any long-term predictions. Their criticism about
the limited predicative capabilities of SIR model is not supported by their results.
The generalized SIR model with proper procedure of its parameter identification
can give rather good log-term prediction for the accumulated and daily numbers of
COVID-19 cases during any pandemic wave(see examples in [*]).
 The conclusions look too bulky. I think the last paragraph could be enough.   [*] I. NESTERUK. SIMULATIONS OF NEW COVID-19 PANDEMIC WAVES  IN UKRAINE AND IN THE WORLD  BY GENERALIZED SIR MODEL. System Research & Information Technologies, 2022, â„– 2, pp. 94-103.  DOI: 10.20535/SRIT.2308-8893.2022.2.07

Author Response

Please, see attached file

Round 2

Reviewer 3 Report

Comments and Suggestions for Authors

I am safisfied with the new version of the manuscript